

# The human touch: a meta-analysis of anthropogenic effects on plant-pollinator interaction networks

Karla López-Vázquez[1], Carlos Lara[2], Pablo Corcuera[3], Citlalli Castillo-Guevara[2] and Mariana Cuautle[2]

[1] Doctorado en Ciencias Biológicas y de la Salud, Universidad Autónoma Metropolitana, Iztapalapa, Ciudad de México, Mexico
[2] Centro de Investigación en Ciencias Biológicas, Universidad Autónoma de Tlaxcala, San Felipe Ixtacuixtla, Tlaxcala, Mexico
[3] Departamento de Biología, Universidad Autónoma Metropolitana, Iztapalapa, Ciudad de México, Mexico

Corresponding author
Carlos Lara,
carlos.lara.rodriguez@gmail.com

## ABSTRACT

**Background:** Anthropogenic activities significantly impact natural ecosystems, leading to alterations in plant and pollinator diversity and abundance. These changes often result in shifts within interacting communities, potentially reshaping the structure of plant-pollinator interaction networks. Given the escalating human footprint on habitats, evaluating the response of these networks to anthropization is critical for devising effective conservation and management strategies.

**Methods:** We conducted a comprehensive review of the plant-pollinator network literature to assess the impact of anthropization on network structure. We assessed network metrics such as nestedness measure based on overlap and decreasing fills (NODF), network specialization ($H_2'$), connectance (C), and modularity (Q) to understand structural changes. Employing a meta-analytical approach, we examined how anthropization activities, such as deforestation, urbanization, habitat fragmentation, agriculture, intentional fires and livestock farming, affect both plant and pollinator richness.

**Results:** We generated a dataset for various metrics of network structure and 36 effect sizes for the meta-analysis, from 38 articles published between 2010 and 2023. Studies assessing the impact of agriculture and fragmentation were well-represented, comprising 68.4% of all studies, with networks involving interacting insects being the most studied taxa. Agriculture and fragmentation reduce nestedness and increase specialization in plant-pollinator networks, while modularity and connectance are mostly not affected. Although our meta-analysis suggests that anthropization decreases richness for both plants and pollinators, there was substantial heterogeneity in this regard among the evaluated studies. The meta-regression analyses helped us determine that the habitat fragment size where the studies were conducted was the primary variable contributing to such heterogeneity.

**Conclusions:** The analysis of human impacts on plant-pollinator networks showed varied effects worldwide. Responses differed among network metrics, signaling nuanced impacts on structure. Activities like agriculture and fragmentation significantly changed ecosystems, reducing species richness in both pollinators and plants, highlighting network vulnerability. Regional differences stressed the need for

tailored conservation. Despite insights, more research is crucial for a complete
understanding of these ecological relationships.

Plant-pollinator networks

# INTRODUCTION

Anthropization, the process by which humans transform natural environments or
ecosystems (*Steffen et al., 2011*), involves a complex interplay of various factors. Primarily,
it entails land use change, encompassing activities such as urbanization, habitat
fragmentation, deforestation, intentional fires, and changes in agricultural practices. These
alterations to the landscape are accompanied by significant environmental consequences,
including soil degradation, and increased pollution (*Foley et al., 2005*; *Vitousek et al.,
1997*). Together, these factors contribute to a multitude of challenges, spanning economic,
political, and environmental domains. The resulting impacts pose substantial threats to
biodiversity such as biotic homogenization (*Sidemo-Holm et al., 2022*), decrease in species
diversity (*Gross et al., 2009*), loss of specialist species (*Auffret et al., 2018*), decrease in
functional diversity (*Tinoco, Santillán & Graham, 2018*), among others. Moreover,
alterations in land use can disrupt crucial interactions between species, such as those
between pollinators and plants (*Kremen et al., 2007*).

Animal pollination is crucial for the sexual reproduction of the majority of flowering
plants (*Kremen, James & Pitts-Singer, 2008*; *Campbell et al., 2012*; *Cardoza, Harris &
Grozinger, 2012*). The efficiency of pollinators in transporting compatible pollen to plant
stigmas profoundly influences reproductive success. Consequently, the decline in
pollinators can trigger adverse impacts on the life cycle of zoogamous plants and lead to
reductions in species populations (*Yao, Holt & Marshall, 1999*; *Lennartsson, 2002*). To
truly grasp the impacts of habitat modification on species survival and community
composition, it is imperative to transcend species richness and recognize that all species
are intricately interconnected by ecological interactions (*Valiente-Banuet et al., 2015*).

Plant-pollinator interactions exist as complex networks, organized into local groups of
plants and pollinators (*Biella et al., 2019*). Thus, an ecological interaction network
approach may help understand the impact of land conversion on both interacting
communities for several reasons. Firstly, ecological interaction networks provide a visual
and quantitative representation of species relationships in a given ecosystem. This enables
the identification of emergent patterns and aids in understanding the structure and
dynamics of biological interactions (*Tylianakis et al., 2008*; *Bascompte & Jordano, 2013*).
Secondly, this approach allows for the assessment of community resilience to changes in
land use. By analyzing how interactions redistribute in response to landscape alterations,
we can better understand the communities' capacity to adapt and persist in the face of
disturbance (*Tylianakis, Tscharntke & Lewis, 2007*). Furthermore, ecological interaction

networks can help identify key species or "keystone species" that play crucial roles in maintaining ecosystem integrity and stability. Understanding how these species interact with their environment and other species can lead to more effective conservation strategies (*Montoya, Pimm & Solé, 2006*).

Studies exploring the impacts of anthropization on plant-pollinator interaction networks have been pivotal in our understanding of how human activities have fundamentally altered these vital ecological relationships (*e.g.*, *Potts et al., 2010*; *Burkle, Marlin & Knight, 2013*; *White, Collier & Stout, 2022*; *Pardee et al., 2023*). The focus of these investigations primarily centers on discerning the shifts in structural patterns within these networks. These changes are driven by an assortment of influences, including urbanization (*Marín-Gómez, Flores & Arizmendi, 2022*; *Marcacci et al., 2023*), fragmentation (*Grass et al., 2018*; *Jauker et al., 2019*), agriculture (*Marrero, Torretta & Medan, 2014*; *Moreira, Boscolo & Viana, 2015*), deforestation (*Newton et al., 2018*; *Muñoz-Galicia et al., 2023*), and intentional fires (*Peralta et al., 2017*; *Banza et al., 2019*) which have been studied extensively in recent years. Importantly, these same factors constitute the primary drivers behind the alarming worldwide decline in pollinators (*Adedoja & Kehinde, 2018*). The sensitivity of pollinators to habitat alterations is reported in many studies, resulting in reductions in species richness and abundance (see *Burkle, Marlin & Knight, 2013*). Moreover, these alterations provoke shifts in species composition and the foraging behavior of pollinators, with far-reaching ecological consequences (*Murcia, 1996*; *Aizen & Feinsinger, 2003*). Nevertheless, it is not a uniform decline, as anthropized ecosystems affect various species differently. Some species suffer harm (*Ewers & Didham, 2006*; *Cardoso & Gonçalves, 2018*), while others might even benefit (*Steffan-Dewenter et al., 2007*; *Cusser, Pechal & Haddad, 2021*).

Overall, land conversion activities can affect richness and abundance of both plant and pollinator species in several ways (*Potts et al., 2010*). For example, urbanization often leads to the loss and fragmentation of natural habitats, resulting in reduced floral resources and nesting sites for pollinators (*Hall et al., 2017*). This habitat loss can negatively affect both plant and pollinator communities, leading to declines in species richness and abundance (*Steffan-Dewenter & Tscharntke, 2003*). Likewise, fragmentation of natural habitats due to urbanization and other land-use changes can disrupt plant-pollinator interactions by reducing connectivity between habitat patches (*Debinski & Holt, 2000*). Fragmented landscapes may limit the movement of pollinators, resulting in decreased pollination services and altered plant reproductive success (*Steffan-Dewenter & Tscharntke, 2003*). Meanwhile, agricultural intensification involves the conversion of natural habitats into croplands, leading to habitat loss and the simplification of landscapes (*Kremen, Williams & Thorp, 2002*). Monoculture farming practices and pesticide use in agriculture can further exacerbate the decline of both plant and pollinator species (*Goulson et al., 2015*). The loss of floral diversity in agricultural landscapes reduces food resources for pollinators, impacting their abundances and diversity (*Klein et al., 2007*). Lastly, intentional fires, often used as a land management tool in agricultural and natural areas, can also impact plant-pollinator interactions. While some fire-adapted species may benefit from fires,

others may suffer due to habitat destruction and changes in resource availability (*Bond & Keeley, 2005*). Fires can alter the composition of plant communities and disrupt pollinator foraging patterns, affecting pollinator abundances and diversity (*Moreira et al., 2011*).

The primary objective of our study was to made a systematic review to synthesize the existing body of literature concerning plant-pollinator interaction networks in anthropized environments. Our emphasis was on understanding the impact of human activities on the structural characteristics reported on these networks through two predictors: network metrics directly related to network structure, and the richness of interacting species. We utilized various network indices, including nestedness, specialization, connectance, and modularity, to conduct our analysis. Additionally, our review aimed to compare species richness of both pollinators and plants on interaction networks studied in both anthropized and conserved environments using a meta-analysis. The richness of the interacting species is essential for maintaining the structure, stability, and functioning of these networks (*Ives, Klug & Gross, 2000*). We assessed whether the effect size considered (*i.e.*, the effect of antropization on plant/pollinator richness) varied depending on factors such as the type of organism (plant or pollinator), the type of disturbance (*i.e.*, urbanization, fragmentation, agriculture, and intentional fires), the continent and climate of the study site, and the size of fragmented areas where the revised studies were carried out. These assessments were conducted through a comprehensive meta-analysis, providing a more detailed understanding of the multifaceted impact of anthropization on plant-pollinator interaction networks.

Based on the observed trends in the reviewed literature, we expect that networks from conserved environments would harbor greater richness and diversity of pollinators and plants compared to anthropized environments. Previous studies have shown that habitat conservation is associated with higher abundance (*Díaz & Cabido, 2001*; *Winfree, Bartomeus & Cariveau, 2011*) and species diversity (*Potts et al., 2010*; *Hall et al., 2017*). Furthermore, we expect that interaction networks between pollinators and plants in anthropic environments would exhibit a decrease on levels of nestedness and modularity, and they should be less specialized (*Tylianakis et al., 2008*; *Bascompte & Jordano, 2013*). These expectations stem from the premise that human-disturbed environments often lead to the loss of specialist plant and pollinator species, as specialized interactions diminish, the network becomes less structured, leading to a decrease in nestedness and modularity within the network (*Colom, Traveset & Stefanescu, 2021*; *Prendergast & Ollerton, 2021*). Likewise, we expect more generalized networks in altered habitats due to the extirpation of rare and less linked species, or the introduction of alien species which outcompete native specialists (*Xiao et al., 2016*). Lastly, higher network connectance values are expect to be found in anthropized environments (*Doré, Fontaine & Thébault, 2021*). This expectation arises from the potential for human activities to introduce novel interactions, thereby increasing the density of connections within the ecological network and leading to higher connectance values (*Valdovinos et al., 2009*).

## MATERIALS AND METHODS

### Search protocol and data collection

A comprehensive literature search was conducted in scientific journals reporting plant-pollinator interaction networks within anthropized environments. From the articles selected through this search, a consultation of the literature cited in them was also performed (referred to as "Additional studies identified through other sources" in Fig. 1). Keyword searches and their combinations were used. These included "interaction networks" AND "pollinators" AND "diversity" AND "fragmentation" AND "urbanization" AND "agriculture" AND "intentional fires" AND "deforestation" AND "livestock farming", OR "land use change", OR "habitat loss", along with the specific terms: "bats", "bees", "beetles", "bumblebees", "birds", "butterflies", "flies", "hoverflies", "ants", "wasps","moths", "hummingbirds", "reptiles" and "mammals". The selected pollinator taxa were chosen based on their known importance as pollinators in natural and anthropized environments. These taxa represent a diverse range of pollinator groups commonly found in human-altered landscapes and play key roles in pollination services for both wild and cultivated plants. For our literature search, both single and multi-taxa networks were included. After this, and based on how pollinators were grouped in the studies found, we decided to conduct the following grouping classification of pollinators for our analyses: (1) "Bees", studies considering only these hymenopteran species; (2) "Bees and others"; studies considering a group of pollinators consisting of species of bees, hoverflies, bumblebees, and wasps; (3) "Butterflies"; studies considering only lepidopteran species; (4) "Insects": encompassing studies considering species from all the aforementioned groups together as pollinators; (5) "Hummingbirds", studies considering only this bird group as pollinators; and (6) "General", studies considering collectively mammals and species from all the aforementioned groups as pollinators. Each of the chosen studies clearly specified the main type of anthropogenic activity assessed (aligning with our search criteria), and none of these considered the interaction of two or more types of disturbance (*e.g.*, habitat loss × fragmentation). Articles were retrieved through an intensive search in the public databases Web of Science and Scopus (final data base-query 20 September 2023). For all publications which we could not access full texts, we attempted once to acquire it by contacting the corresponding author, otherwise, the study was excluded.

Selection criteria: After conducting the literature search, article titles and abstracts were reviewed to determine whether they met the inclusion criteria for the review. Subsequently, articles meeting these criteria were read in their entirety. After this, only articles published from 2010 to 2023 met our selection criteria, and therefore were chosen for the analyses. Only studies that assessed the impact of anthropogenic activities on the structural patterns of plant-pollinator interaction networks, including both binary and quantitative networks, allowing for comparisons of pollinator and plant species richness within these networks, and studies considering a minimum of three sampling sites for each management class (conserved and anthropized) were included (see Fig. 1). Likewise, we only selected studies that measured network metrics between a conserved site and an anthropized site; all

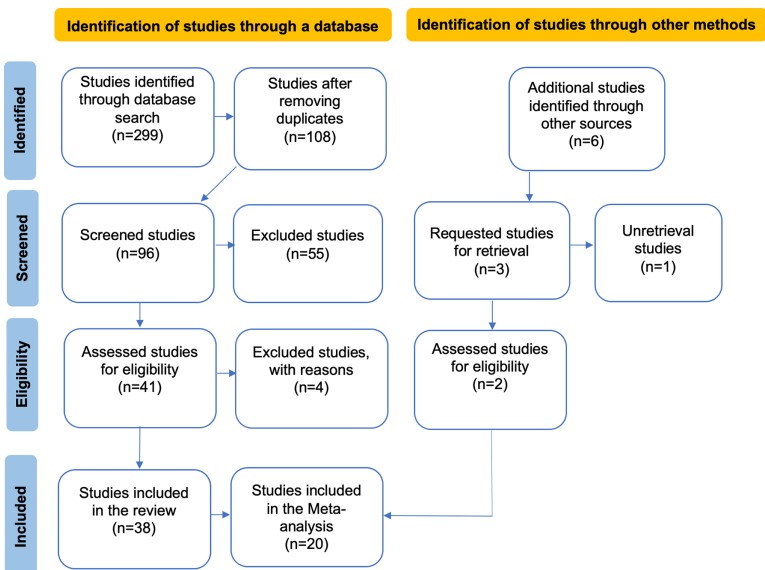

**Figure 1 PRISMA flow diagram illustrating the data compilation process for the systematic review and meta-analysis.** In the flowchart, the articles obtained from other sources are a result of reviewing the reference lists of the articles identified in the database search, while the excluded articles are those that did not meet the evaluation criteria in our review.

studies not meeting these criteria were excluded. Only one of the reviewed studies analyzed five interaction networks with different taxa as pollinators, providing the necessary information for conducting our analyses for each network (*i.e.*, *Weiner et al., 2011*). Therefore, only this study was considered as five separate studies. Although other articles also analyzed multi-taxa networks, the specific information required for their division was not provided.

In order to use certain metrics in meta-analyses, it must be possible to calculate the standard error (*Harrer et al., 2021*). For this reason, we only included studies that reported the standard error or provided enough information to calculate it. Additionally, studies with at least three repetitions for each management class mentioned above were included to ensure robustness in effect size estimation. Sampling effort was not considered as a selection criterion because, in a meta-analysis, each effect size is weighted by its pooled standard error, thus accounting for variations that may arise from different sampling efforts (*Harrer et al., 2021*). By adhering to these explicit inclusion and exclusion criteria, potential disagreements were resolved. In cases where discrepancies persisted, all co-authors engaged in thorough discussions and reached a consensus to understand each other's perspectives and arguments regarding the inclusion or exclusion of specific studies in our analyses (*Uttley et al., 2023*).

## Network properties and their metrics considered in the analyses

To describe the impact of anthropization on the structure of interaction networks between plants and pollinators, we included only studies that describe the following network properties: nestedness, network specialization, connectance, and modularity, using specific

metrics (*i.e.*, NODF, $H_2$', C and Q, respectively) for our analyses. These network-level parameters facilitate a comprehensive understanding of the overall interaction structure within the community (*Dormann et al., 2009*). Widely recognized and commonly used, these measures quantify key aspects of plant-pollinator interaction networks, ensuring comparability and consistency with existing literature.

Nestedness refers to a pattern where the interactions among species or nodes in a network are structured in such a way that the interactions of specialists are subsets of the interactions of generalists (*Jordano, Bascompte & Olesen, 2003*; *Guimarães et al., 2006*). The nestedness pattern holds significant ecological importance as it acts as a kind of insurance for the long-term functioning of the ecosystem. It serves as a buffering mechanism against environmental variations (*Thébault & Fontaine, 2010*) and significantly contributes to the stability of such networks (*Bastolla et al., 2009*; *Thébault & Fontaine, 2010*) by promoting greater resilience to extinction for well-connected generalist species (*Aizen, Sabatino & Tylianakis, 2012*). We used the nestedness metric based on overlap and decreasing fill (NODF), which is a commonly used measure to quantify nestedness in interaction networks. NODF calculates the extent to which the interactions of specialist species are nested within the interactions of more generalist species. It considers both the presence and absence of interactions between species and ranges from 0 to 100, where higher values indicate greater nestedness (*Almeida-Neto et al., 2008*). Anthropization can impact nestedness properties in plant-pollinator interaction networks due to changes in habitat structure and species composition. Therefore, exploring this metric allows us to infer their vulnerability or resilience to these human pressures.

Connectance refers to the proportion of potential interactions that are realized in a network. It measures the density of interactions between species or nodes in a network, indicating how well-connected or dense the network is. The connectance metric (C) quantifies this proportion by dividing the number of observed interactions by the total number of possible interactions in the network. It typically ranges from 0 to 1, where a value of one indicates that all possible interactions are realized, while a value of zero indicates that Io interactions are observed (*Dormann et al., 2009*). In our study, measuring connectance helps us assess how human activities, which alter habitat structure, floral resource availability, and pollinator abundance, can impact the density and efficiency of interactions within the plant-pollinator networks (*Tavares-Brancher et al., 2024*).

Network-level of specialization is $H_2$', which quantifies the degree of specialization in a bipartite ecological network, such as plant-pollinator interaction networks. Is calculated based on the relative abundance of interactions between species, considering both the number of partners and their strengths of interaction (*Blüthgen, Menzel & Blüthgen, 2006*). Values of this metric ranges between 0 and 1. A value of zero indicates a completely generalized network, where all species interact with all others equally. This suggests low specialization. A value of one indicates a completely specialized network, where each species interacts exclusively with one partner. This suggests high specialization. By measuring this metric, we can infer whether anthropic activities promote generalized or specialized networks. For example, habitat conversion often results in networks becoming less specialized. This occurs as rare and less connected species disappear, generalist species

become more abundant, and many pollinator species exhibit more generalized behavioral traits (*Xiao et al., 2016*).

Modularity is a parameter that indicates whether there are two or several groups of individuals or species that have more interactions among themselves than with other groups in the network (*Marquitti et al., 2014*). Metric Q is a quantitative measure used to assess the degree of modularity in a network. It quantifies the difference between the observed number of intra-module edges and the expected number of such edges in a randomized null model. A higher Q value indicates a higher level of modularity. The range of Q values typically falls between −1 and 1. A positive Q value indicates that the network is more modular than expected by chance, while a negative Q value suggests that the network is less modular than expected. Q values close to zero indicate that the network has a similar level of modularity as random networks (*Olesen et al., 2007*). We assessed modularity in our study because it has been observed that the modularity of networks increases their robustness against disturbances, as specialized interactions are concentrated within the modules and do not affect other species outside of those modules (*Stouffer & Bascompte, 2011*). Thus, A higher modularity value (Q) suggests a stronger partitioning of species interactions within distinct ecological communities, which could indicate a more stable and specialized network structure, commonly found in conserved environments (*Montoya, Pimm & Solé, 2006*). Conversely, a lower value indicates a network with weaker or fewer modular divisions, suggesting a more interconnected and potentially less stable network structure, which may be characteristic of disturbed environments (*Rodewald et al., 2014*).

## Statistical analyses

A synthesis was conducted on the potential impact of anthropization on the structural patterns of plant-pollinator interaction networks, describing the decrease, increase, or lack of effect as reported by the authors. As an initial step in assessing the impact of anthropogenic activities on the structure of plant-pollinator interaction networks, we examined variations in metric values associated with network structural patterns, such as NODF, $H_2$', connectance, and modularity (as response variables), across different anthropogenic activities, including deforestation, urbanization, fragmentation, agriculture, and livestock (as a fixed factor), using analysis of variance (ANOVA). To identify specific groups of anthropogenic activities that displayed significant differences in their network metric values, we utilized Tukey *post hoc* tests. These tests were performed using the R Studio software (*RStudio Team, 2020*).

## Effect size calculation and simple meta-analysis

To conduct a meta-analysis, it was necessary to determine an effect size that could be summarized across all studies. In this regard, the effect size is defined as a metric that quantifies the relationship between two entities, capturing the direction and magnitude of this relationship (*Harrer et al., 2021*). In this case, the chosen effect size indicated the impact of anthropization on the richness of pollinators and plants. The effect size was selected using the standardized mean difference (SMD), with the Hedges' g correction for

small samples (*Hedges & Vevea, 1996*). This was because the means of plant and pollinator richness between conserved and anthropized environments were the most frequently reported response variables in the collected articles. The calculation of SMD for each study involved subtracting the mean of pollinator or plant richness from the conserved site from the mean of pollinators and plants from the anthropized site in each respective study, and then dividing this difference by the pooled standard deviation (*Harrer et al., 2021*). Positive values indicate a higher number of pollinators or plants in the preserved site, while negative values indicate the opposite. A value of 0 represents the absence of an effect size. When means and standard deviations were not reported in an article, other reported statistics that could be converted to SMD, such as correlation coefficients, chi-square ($\chi^2$), one-way ANOVA, and two-sample t-test, along with their corresponding formulas, were used (*Harrer et al., 2021*). In cases where a document did not provide any of these data, it was excluded from the meta-analysis. This simple meta-analysis comprised $K = 36$ studies because some articles included the richness for different pollinator taxa or both groups (plants and pollinators), allowing us to obtain more than one SMD from certain publications. This practice is common in meta-analyses as long as the effect size obtained aligns with what was specified in its definition (*Borenstein et al., 2009*).

For this meta-analysis, a random-effects model was employed. This model assumes that studies do not reflect a single true effect due to differences in populations, interventions, comparators, or outcome assessment methods (*Fernández-Chinguel et al., 2019*). This approach allowed the comparison of results across different studies, even when they did not measure the parameters of interest in the same way. The maximum likelihood (ML) method was used to obtain tau$^2$, which measures the variability between the effects of different studies (*Higgins, 2011*), and the Jackson method was used to calculate confidence intervals for tau$^2$ and tau (*Borenstein et al., 2009*). We used Knapp-Hartung adjustments (*Knapp & Hartung, 2003*) to calculate the confidence interval around the summary effect. The Knapp-Hartung adjustment aims to control for the uncertainty in the estimate of between-study heterogeneity and is based on a *t* distribution. A *t*-test was employed to determine the significance of the summary effect size. To assess the heterogeneity of effect sizes, we utilized *Q* statistics (*Hedges & Olkin, 1985*), which represent weighted sums of squares following an approximately asymptotic chi-square distribution. These statistics facilitate various tests; in this instance, we evaluated whether the variance among effect sizes exceeded what would be expected by chance (*Cooper, 1998*). Also, we calculated the $I^2$ statistic, which determines the percentage of the total variability in a set of effect sizes due to true heterogeneity, that is, the between-study variance (*Gurevitch & Nakagawa, 2015*). All analyses were conducted using the dmetar (*Harrer et al., 2019*), meta (*Balduzzi, Rücker & Schwarzer, 2019*) and metafor packages (*Viechtbauer, 2010*) in the RStudio software (*RStudio Team, 2020*), and the results are displayed in forest plots.

## Subgroup analysis

Additionally, subgroup analysis was performed to determine if the pattern of heterogeneity in effect size was related to the group of organisms evaluated (pollinators and plants), climate, continent, or anthropogenic activity. For the climate subgroup, climate types

followed the *Köppen (1923)* climate classification system, which divides the Earth into five main climate categories (A = tropical rain climate without cool season, B = dry climates, C = warm temperate climates, D = snow climates and E = ice climates) and with further sub-divisions according to the seasonal distribution and amount of rainfall, and to the winter and summer temperature regimes (f = sufficient moisture in all months; s = dry season in the summer of the respective hemisphere; w = dry season in the winter of the respective hemisphere). In each utilized article, the climate at the sampling site was categorized following this classification. Due to the limited number of studies in some subgroups, we followed the recommendation of *Borenstein et al. (2009)* to use a single tau$^2$ estimate for all subgroups in our analysis. This approach allows for a more stable and reliable estimate of between-study variance, particularly when some subgroups have few studies.

Subgroup analysis depends on statistical power, so it only makes sense when the meta-analysis contains at least $k = 10$ studies. In our case, the number of studies in our meta-analysis was $k = 36$. Only one study included in the review examined livestock as an anthropogenic activity; therefore, this activity was not included as a category in the analyses.

The model for subgroup analyses is a mixed-effects model, because contains both random effects (within subgroups) and fixed effects (since subgroups are assumed to be fixed). The $Q$-test was used to compare the observed value to the expected value (*i.e.*, residual heterogeneity, QE), assuming a chi-squared ($\chi 2$) distribution with degrees of freedom G-1, where G is the number of groups. If the observed value was significantly greater than the expected value, the $p$-value from the test indicated the presence of a real difference in effect sizes among subgroups (*Higgins, 2011*). Also, we calculated the $I^2$ statistic for each subgroup.

### Meta-regression

Furthermore, a meta-regression was conducted to identify specific continuous variables explaining heterogeneity between studies (*Borenstein et al., 2009*; *Koricheva, Gurevitch & Mengersen, 2013*). In this type of analysis, one or more predictor variables can be used to predict real differences in effect sizes, considering mixed-effects models (*Higgins, 2011*). In our study, we only considered the size of the habitat fragments (total coverage in square meters) where the studies were conducted as predictor variables. Comparison between the full model and the reduced model was performed using the likelihood ratio test. If the full model proved superior to the reduced model, the fragment size variable was retained. The Knapp-Hartung adjustment was used to obtain more robust estimators (*Higgins, 2011*).

### RESULTS

Network metrics from a total of 38 articles published between 2010 and 2023, were use for the ANOVAs (see Supplemental File 1). The total number of studies compiled per continent was: Africa (four), the Americas (15), Asia (two), Europe (16) and, Oceania (one). The most represented anthropogenic activities included agriculture (15 studies) and fragmentation (11 studies), while the most well-represented taxon was that of insects, with

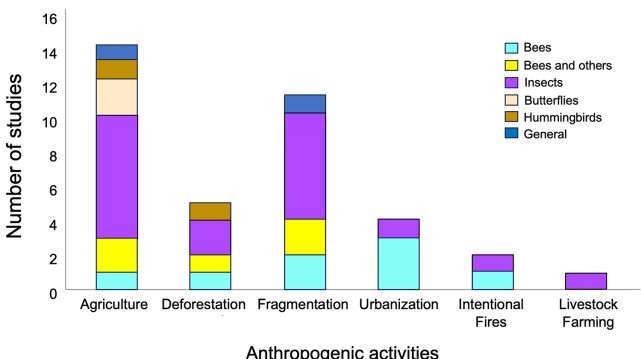

**Figure 2 Representativeness of papers by taxonomic group in studies on plant-pollinator networks conducted in locations with anthropogenic activities, which were used in the meta-analysis.** Taxonomic groups were categorized as follows: (1) "Bees," referring to studies considering only these hymenopteran species; (2) "Bees and others," which includes studies considering a group of pollinators consisting of species of bees, hoverflies, bumblebees, and wasps; (3) "Butterflies," focusing on studies considering only lepidopteran species; (4) "Insects," encompassing studies considering species from all the aforementioned groups together as pollinators; (5) "Hummingbirds", covering studies considering only this bird group as pollinators; and (6) "General," incorporating studies considering collectively mammals and species from all the aforementioned groups as pollinators .

33 studies (Fig. 2, Supplemental File 1). Likewise, the effects observed in the various metrics of plant-pollinator interaction networks in response to anthropogenic activities exhibited highly variability (see Table S1 in Supplemental File 2).

Anthropogenic activities have varying effects on metrics associated with the structure of plant-pollinator interaction networks (Fig. 3). NODF values showed significant variation among different anthropogenic activities in the studies we evaluated ($F = 16.15$, $d.f. = 2$, $p = 0.007$), with agriculture ($p = 0.001$) and fragmentation ($p = 0.009$) being the primary determining factors. A similar significant effect was observed for network specialization ($H_2$') ($F = 0.02$, $d.f. = 3$, $p = 0.02$), where deforestation ($p = 0.02$) and fragmentation ($p = 0.03$) contributed to these differences. Conversely, no significant differences were found among anthropogenic activities in their effects on connectance values ($F = 0.9$, $d.f. = 3$, $p = 0.46$). Regarding modularity, ANOVA was not applicable due to the limited sample size of the reviewed studies and the lack of variance homogeneity.

## Meta-analysis

For the meta-analyses, only 20 out of the 38 studies met the inclusion criteria, and for the final dataset used, 36 effect sizes were included (16 for plant species and 20 for pollinator species).

The average effect size was 0.52, with a 95% confidence interval ranging from 0.07 to 0.96. The associated $p$-value was significant ($t = 2.37$, $d.f. = 35$, $p = 0.02$), indicating that anthropogenic disturbance reduces the richness of pollinators and plants (see Fig. 4). The value of tau$^2$ was 1.21, while the $I^2$ value was 83.8%. The test of heterogeneity was significant ($Q = 215.75$, $d.f. = 35$, $p < 0.0001$), suggesting variability in effect sizes among different studies. The results indicated a statistically significant difference in the richness of

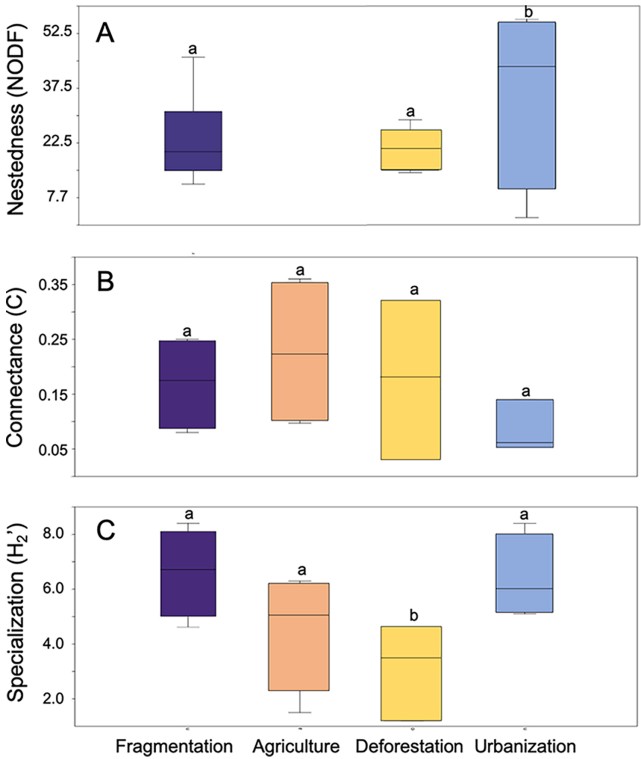

**Figure 3 Box plot representing the variation in network metrics in plant-pollinator networks assessed at study sites under different anthropic activities.** (A) Nestedness metric based on overlap and decreasing fill, (B) Network connectance, and (C) Network speciali. Different letters denote significant differences among anthropogenic activities (*p*-value < 0.05) after Tukey's correction for multiple comparisons.

pollinator and plant species between conserved and anthropized sites. However, there is also significant heterogeneity among the studies. This suggests that the actual effect may vary depending on the type of disturbance or that additional factors, beyond anthropization, may influence species richness of pollinators at different sites (see Table S2 in Supplemental File 2, Fig. 4).

## Subgroup analysis

A subgroup analysis revealed significant differences in the observed effects within subgroups of pollinators, plants, continents, and anthropogenic activities and climate subgroup (see Table S2, Fig. 5).

Pollinators and Plants. The group of 'Pollinators' showed the largest effect size ($g = 0.13$) compared to 'Plants' ($g = 0.79$; Table S2). The tau$^2$ value for 'Pollinators' ($\tau^2 = 0.88$) indicates low heterogeneity, meaning that the studies regarding this group are consistent in their findings. On the other hand, the tau$^2$ value for 'Plants' ($\tau^2 = 1.37$) indicates a moderate level of heterogeneity, suggesting that there is some variability in the effect sizes reported in the studies concerning this group. This variability may be due to differences in study characteristics, methodologies, or other factors that influence the relationship between 'Pollinators' and the structure of the studied networks.

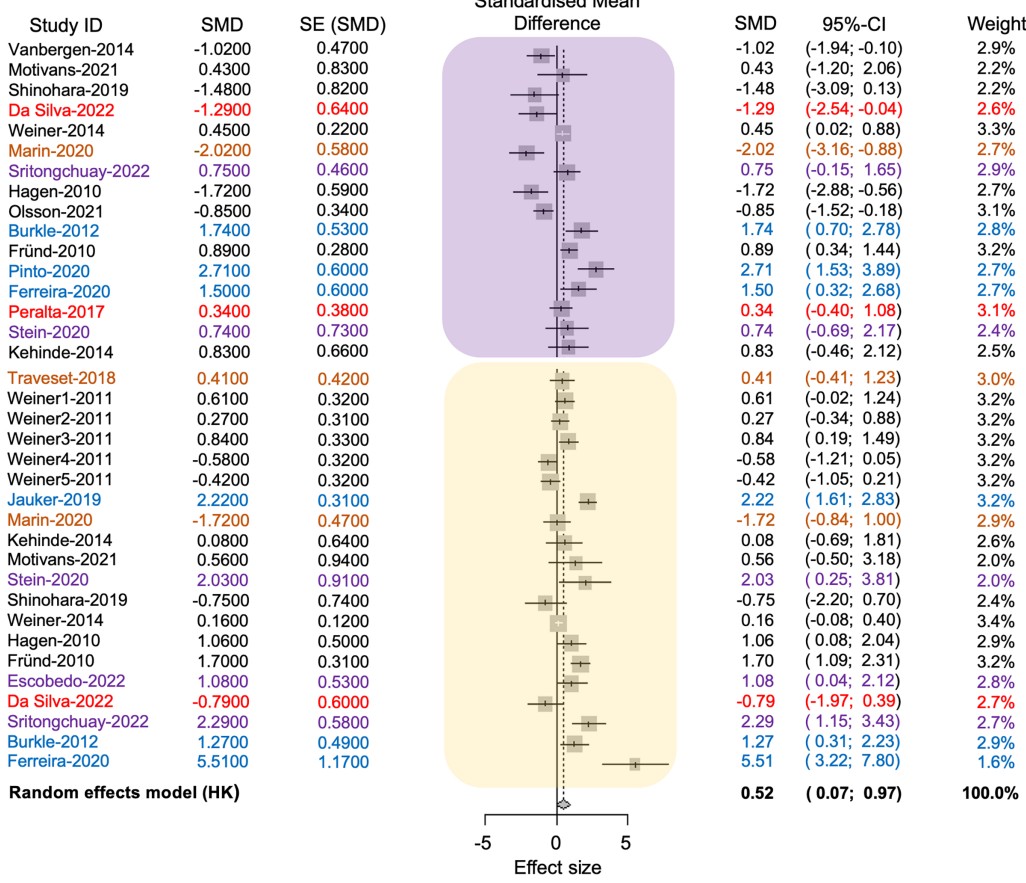

**Figure 4 Forest plot displaying the effect size and 95% confidence intervals (CI,) represented by vertical lines, for pollinator (embedded in the yellow square) and plant (embedded in the purple square) richness.** The plot presents measurements in conserved (positive effect size values) and disturbed sites (negative effect size values) across each study (Study ID). References and their values are color-coded according to the anthropogenic activities where the plant-pollinator networks were studied as follows: black = agriculture, red = intentional fires, brown = urbanization, purple = deforestation, blue = fragmentation.                           

Anthropogenic activity. Our results indicate that 'Fragmentation' have the most higher size effect ($g = 2.07$, Table S2) on the plant/pollinator richness within the anthropogenic activity category. The high SMD value suggest that this activity have a substantial impact on the richness of plant-pollinator networks, and the low tau$^2$ value ($\tau^2 = 0.11$, Table S2) indicate that this effect was consistent among the different studies within this subgroup. We can also observe that the other types of anthropogenic disturbances had similar negative effects in plant and pollinators richness (Table S2).

Continent. Our results indicate differences in effect sizes and levels of heterogeneity among continents. 'America' shows the largest effect size ($g = 0.66$, Table S2) with moderate between-study heterogeneity ($\tau^2 = 2.64$, Table S2). However, 'Asia', 'Africa' and 'Europe' exhibit similar lower effect sizes ($g = 0.5$ in all cases) with moderate levels of heterogeneity ($\tau^2 = 1.59$, $\tau^2 = 0.84$ and $\tau^2 = 0.57$, respectively; see Table S2).

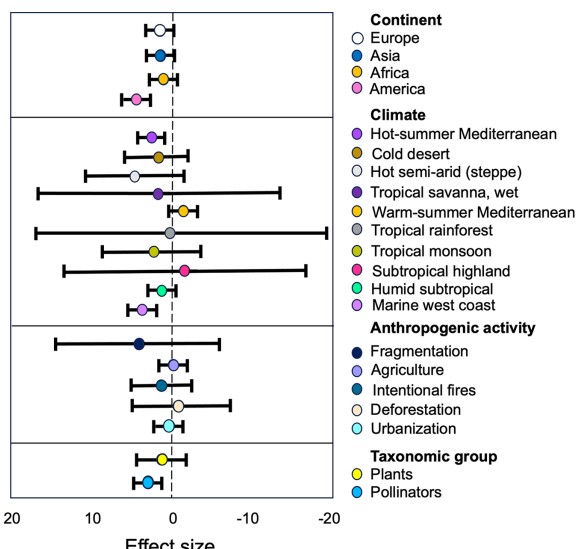

**Figure 5 Estimates of the strength of the relationships between anthropization, classified as subgroups of moderators, and the structure of plant-pollinator networks.** The forest plot illustrates how the anthropization effect varies based on the continent, climate, type of anthropogenic activity, and taxonomic group. The points represent mean effect size estimates from the models, while the lines represent 95% confidence intervals. Positive values (left) indicate a higher plant/pollinator richness in conserved sites than in anthropized.

Climate. The subgroup analysis of the climate category, based on the Köppen climate classification system, reveals significant differences ($p = 0.0001$) in the observed effects of anthropogenic activities on pollinators and plants across various climate types. The hot semi-arid (steppe) climate and the tropical savanna, wet climate exhibit the highest standardized mean differences (SMD) at 2.04 and 1.85, respectively, suggesting that these dry and hot climates may be more susceptible compared to cooler and wetter climates to the impacts of human activities. However, the wide 95% confidence intervals for these estimates indicate considerable uncertainty (see Table S2).

## Meta-regression

The model used in this analysis exhibited a significant amount of residual heterogeneity that is explained by the fragment size variable ($QE = 180.53$, $d.f. = 33$, $p < 0.0001$).

## DISCUSSION

As expected, anthropogenic activities significantly impact plant-pollinator interaction networks, influencing network metrics and species richness, with insects being the most represented taxon. Fragmentation and agriculture are key drivers of nestedness variation, and network specialization is mainly influenced by deforestation, while different anthropogenic activities showed no significant impact on connectance values. Our findings indicate reduced species richness in response to human disturbance, particularly due to fragmentation. Heterogeneity among studies suggests varied responses to anthropogenic disturbance or additional factors. Heterogeneity was observed among continents, with 'America' showing the largest effect size and moderate heterogeneity. Climate also

influenced effect sizes, with dry and hot climates exhibiting the largest effect size and slightly higher heterogeneity compared to cooler and wetter climates. Lastly, fragment size is as a significant predictor of residual heterogeneity, emphasizing the influence of landscape-scale factors on the response of plant-pollinator network richness to anthropogenic disturbance. Below, we discuss our findings, providing an overview of how plant-pollinator networks could be affected by anthropogenic activities, highlighting the consideration of other factors to explain these impacts, and suggesting future directions.

Our literature review revealed that the effects of anthropization have been mainly studied in plant-insect networks. While birds, mammals, and reptiles contribute to the pollination of agricultural crops and wild plants, insects, particularly bees, are the primary pollinators (*Potts et al., 2010*). For instance, approximately 70% of the 124 major crops consumed worldwide by humans depends on insect pollinators (*Klein et al., 2007*). The abundance, diversity, functional importance, and economic significance of insects, coupled with methodological ease, have driven the focus on plant-insect networks in the literature (*Petsopoulos et al., 2021*). However, bird species such as hummingbirds (Trochilidae) in the Americas, sunbirds and spiderhunters (Nectariniidae) in Africa and Asia, and honeyeaters (Meliphagidae) in Australasia also play significant roles as pollinators (*Cronk & Ojeda, 2008*). Among mammals, bats are the principal pollinators, pollinating numerous economically and ecologically important plants that provide valuable products to humans; and some reptiles are known to have important pollination roles (*e.g.*, *Olesen & Valido, 2003*). While existing research has predominantly focused on plant-insect networks, there is a clear imperative for further exploration that integrates these less-studied pollinator groups within the broader context of human-induced impacts on pollination networks. Such efforts are crucial for elucidating the generalizability of observed impacts and advancing our comprehension of ecosystem dynamics amidst anthropogenic pressures.

Identifying the structural patterns most susceptible to alterations caused by anthropogenic activities in the plant-pollinator interaction networks helps to understand the dynamics of mutualistic interactions and the vulnerability of floral visitors. Our results showed that agriculture and fragmentation are the most impactful activities in the network structure in this research. These activities are widely practiced worldwide. Some articles have examined how these agricultural activities affect plants and their pollinators. They explain how intensified agriculture can influence the availability of native pollinators for crops, affecting food production (*e.g.*, *Kremen et al., 2004*; *Klein et al., 2007*). This highlights that greater expanses of natural habitats contribute to increased stability and predictability of pollination services (*Tscharntke et al., 2005*; *Sardiñas & Kremen, 2014*).

Furthermore, it has been observed that agricultural practices can affect pollinator populations, such as bees (*Raven & Wagner, 2021*), a widely studied taxonomic group in plant-pollinator interaction networks. It has been noted how monocultures and pesticide use impact not only populations of wild bees but also other pollinators (*Roulston & Goodell, 2011*; *Garibaldi et al., 2011*). It is worth mentioning that a significant transformation of natural ecosystems into agricultural land is projected to reach 109 million hectares by 2050.

Although our study did not find significant effects of activities such as livestock farming, urbanization, deforestation, and intentional fires on interaction network metrics, it's important to consider that these results may be due to the study's limitation in focusing solely on structural patterns of networks. However, other studies suggests that these activities also contribute to climate change, which can affect plants through alterations in temperatures, precipitation patterns, and atmospheric carbon dioxide levels, favoring invasive species over native ones (*Dukes & Mooney, 1999*). It has also been examined that when analyzing livestock farming independently from agriculture, how livestock impact pollinator visitation frequency can vary and is further modified by changes in vegetation cover caused by livestock presence (*Tadey, 2008*).

Regardless of the specific factors driving changes in land use, they invariably lead to habitat alterations that can negatively impact pollinator populations (*Winfree, Bartomeus & Cariveau, 2011*). These alterations may include habitat loss, fragmentation, degradation, and the introduction of non-native species, all of which can disrupt pollinator communities, decrease floral resources, and impair pollination services. Therefore, even though changes in land use may vary in intensity, their overall effect on pollinator habitats remains consistently detrimental (*Lázaro & Tur, 2018*). Regarding the impact of intentional fires on pollinators, most studies are limited to comparing burned areas with unburned areas (*Carbone et al., 2019*). The overall results of the two analyzed studies indicated a positive effect on pollinator richness. However, a slight impact on Lepidoptera richness was identified (*Peralta et al., 2017*; *da Silva Goldas et al., 2022*). Regarding the effects of anthropization on the structural patterns of interaction networks, previous research supports our results, concluding that the metrics used are particularly sensitive to environmental changes (*Aguilar et al., 2009*; *Ferreira, Boscolo & Viana, 2013*; *Soares, Ferreira & Lopes, 2017*). However, contrary to our findings, other studies have indicated that empirical data available suggest that nestedness is not affected by habitat disturbance (*e.g.*, *Hagen & Kraemer, 2010*; *Jauker et al., 2019*; *Morrison & Dirzo, 2020*; *Escobedo-Kenefic et al., 2022*). Instead, $H_2$' metrics show responses similar to those obtained in this study. As nested pollination networks are primarily explained by species traits (*Olito & Fox, 2015*) and relative abundance (*Vázquez et al., 2005*), other factors such as resource availability (*Pardee et al., 2023*), community structure (*Dupont, Hansen & Olesen, 2003*), and climate change (*Dalsgaard et al., 2013*) are also important. Therefore, they should be considered in future studies assessing the nestedness of these networks in anthropized environments. Furthermore, changes in the roles of species are also described, with oscillations between generalists and specialists in different conditions. This is because in situations of lower environmental quality, specialist species with specific morphological and behavioral adaptations tend to be lost, as observed in previous research (*Ferreira, Boscolo & Viana, 2013*).

The reviewed studies indicate that, although network nestedness exhibited different effects on anthropogenic activities, we can infer that the explanation for the fourteen studies where nestedness tends to decrease is that authors have observed that any disturbance resulting from anthropogenic activities affects total species richness and the abundance of interactors, which decreases with habitat loss (*Spiesman & Inouye, 2013*).

This is because a reduction in species richness decreases the network size, and in turn, the number of interactions (links) by generalist species (*Olesen et al., 2007*). This is also related to specialization results ($H_2'$), where an increase was observed in most studies. In some cases, no effect was recorded concerning the different impacts of anthropogenic activities. This can be attributed to the loss of specialist species, the increase in generalist species, and the decrease of pollinators specialized in plants sensitive to environmental changes (*Weiner et al., 2014*). Furthermore, this is confirmed by the analysis of variance, which shows that there are indeed significant differences between anthropogenic activities concerning this metric, as it demonstrates that specialization increases in studies where agricultural activities are present.

Consequently, these changes can reduce network robustness, which is less robust or more susceptible when core species are extinguished due to potential alterations in the species that comprise it, resulting in an imbalance in communities (*Tylianakis et al., 2010*). Furthermore, some articles argue that reduced nestedness may be due to the reduction of lower-quality environmental areas (*Burkle, Marlin & Knight, 2013*; *Vanbergen et al., 2014*; *Moreira, Boscolo & Viana, 2015*). An example of this is the observation of a rapid decrease in species diversity over a short period, which primarily affects specialist species and leads to a narrowing of the niche for the remaining generalist species (*Burkle, Marlin & Knight, 2013*; *Moreira, Boscolo & Viana, 2015*). However, in the long term, an even more intense reduction in diversity can be observed, also impacting generalist species (*Burkle & Knight, 2012*). Furthermore, the increase and lack of effect on nestedness in the 11 presented studies could be linked to the concentration of interactions by generalist species, both in plants and pollinators (*Jauker et al., 2019*; *Díaz Infante, Lara & Arizmendi, 2020*; *Morrison & Dirzo, 2020*; *Motivans Švara et al., 2021*; *Escobedo-Kenefic et al., 2022*).

Regarding network connectance, it did not generally change, although it increased in seven studies and decreased in eight. While we may consider that network connectance is not significantly affected in most studies, it's essential to note that some research has found large, highly connected networks in agricultural areas due to the presence of flowering herbaceous plants and fruit trees (*Aavik et al., 2008*). In this regard, some studies suggest that conservation efforts should focus on preserving highly connected communities, seeking empirical evidence of a relationship between connectance (complexity) and the conservation value of communities at different stages of degradation (*Prendergast & Ollerton, 2021*).

As for modularity, the results showed an increase, likely because interaction networks in fragmented sites tend to exhibit modular patterns. This is due to the high specialization in these sites, given the limited number of interacting species (*Santamaría et al., 2018*; *Morrison & Dirzo, 2020*; *Librán-Embid et al., 2021*). However, this modularity could result from a temporal relationship influenced by species phenology (*Morente-López et al., 2018*; *Lázaro & Gómez-Martínez, 2022*). As a result, the meta-analysis revealed an impact on the richness of interacting species, including both plants and pollinators. While several previous studies have evaluated the effects of these anthropic activities (*e.g.*, *Peralta et al., 2017*; *Morrison & Dirzo, 2020*; *Librán-Embid et al., 2021*), none have specifically focused on describing changes in species richness within interaction networks. Furthermore, these

studies have mainly been limited to the group of bees or other insects, with a tendency to increase modularity in their pollination networks.

Given the lack of control over variations in network size among the plant-pollinator networks examined in our study, it is imperative to approach the interpretation of these results with a nuanced understanding of how network size can impact metrics like connectance and modularity. Recognizing the intricate interplay between network size and these metrics is pivotal for drawing precise ecological conclusions and making valid comparisons across networks of different sizes (*Olesen & Jordano, 2002*). While our study offers valuable insights into the diverse interactions between pollinators and plants under various human activities, limitations stemming from the influence of network size emphasize the need for cautious interpretation. Future research efforts should prioritize thorough and meticulous consideration of the potential effects of network size to ensure the validation and refinement of these findings.

Our results highlight that specific anthropogenic activities such as fragmentation, agriculture, and deforestation have the greatest negative effects on both plant and pollinator richness, as well as on metrics associated with the structure of their interaction networks. These activities are extensively performed in many parts of the world where there is very little natural habitat left (*Winfree et al., 2009*). In the context of ecological interaction networks, studies have shown a positive association between the richness of plant species and pollinator species in anthropized areas (*Kral-O'Brien et al., 2021*). Specifically, as the richness of plant species increases, there is also an observed increase in pollinator species richness. However, it's imperative to exercise caution when interpreting this relationship within the network framework. While an increase in species richness may appear beneficial at first glance, the complexity of network metrics necessitates careful consideration. Changes in the network structure, resulting from increases in richness, may have unintended consequences for specialist species or key species within the network, potentially disrupting important ecological interactions (*Vanbergen et al., 2017*). These findings may also be relevant to the results obtained in the meta-regression analysis, where we explored the influence of fragment size on the effect size. We found that this variable did yield significant results. In this context, the composition of vegetation cover in the fragments confirms that having a greater number of flowering species can attract more pollinators (*Herrera, 1987*). Thus, an increase in the richness of both plants and pollinators may occur due, for example, to the colonization of generalist species from the matrix and to edge effects (*Battisti, 2003*). However, as cautioned above, the gradual decline of more sensitive species, caused by changes in the extinction/colonization rates and the proportional increase of egde/generalist species, may induce species turnover in fragments and cascade effects (*Pimm, 1986*).

Our results reveal discrepancies in effect sizes and levels of heterogeneity across continents regarding the influence of anthropization on the richness of pollinators and plants in interaction networks. Specifically, our analysis demonstrates that 'America' exhibits the most substantial effect size, indicating a pronounced impact of anthropogenic activities on pollinator and plant diversity compared to other continents. Conversely, 'Asia', 'Africa', and 'Europe' show similar, albeit lower effect sizes, suggesting a

comparatively weaker influence of anthropization on species richness in these regions. However, moderate levels of heterogeneity within each continent underscore the variability in responses observed among individual studies. This aligns with an ambitious evaluation of the drivers and risks for pollinator declines worldwide. In their study, *Dicks et al. (2021)* used a high-ranking global risk to humans based on reduced species diversity promoted by anthropic activities. Their results showed that the region with most serious risk on the impact of pollinator decline on wild plants and fruits was Latin America, followed by in Africa, and Asia-Pacific. Moreover, most of the studies we considered in our analyses focused on bees, an insect group experiencing declining in North America and Europe (*Moritz & Erler, 2016*). These findings underscore the nuanced geographic patterns in the response of pollinators and plants to anthropogenic pressures, highlighting the importance of considering regional context in conservation efforts (*Hall & Steiner, 2019*). Further investigations are needed to elucidate the underlying mechanisms driving these continental differences and their implications for biodiversity conservation.

Finally, our results indicated a larger effect size in studies conducted in dry and hot climates such as the hot semi-arid (steppe) climate and the tropical savanna, wet climate, compared to those undertaken in cooler and wetter climates like the warm-summer Mediterranean climate and the marine west coast climate. Dry climates typically have lower overall plant productivity and floral abundance (*Kuppler et al., 2021*), and water availability is a limiting factor affecting both plants and pollinators (*Kuppler & Kotowska, 2021*). Therefore, some of the most homogeneous pollinator communities (favoring species with tolerance for higher temperatures and dry conditions) occur in regions with warm temperatures and low precipitation, and anthropization accelerates this process (*Williams & Newbold, 2020*). Despite the fact that most plant-pollination networks have been sampled in temperate zones so far (*Vizentin-Bugoni et al., 2018*), such as the humid subtropical climate and the subtropical highland climate, and tropical regions like the tropical rainforest climate have been less studied despite their greater diversity (*Dáttilo & Rico-Gray, 2018*), our findings highlight the vulnerability of plant-pollinator networks in dry and hot climates to human-induced changes, a topic that deserves further research.

## CONCLUSIONS

An in-depth analysis of patterns in plant-pollinator interaction networks shows a wide range of responses to human activities. The study suggests that intensified agriculture and habitat fragmentation are significant factors harming biodiversity and species interactions. While our results didn't reveal major effects from activities like livestock farming, urbanization, deforestation, or intentional fires, It is possible these impacts are underestimated because we only focused on network structural patterns and richness of the interacting species. Our findings suggest that metrics such as nestedness, $H_2$', connectance, and modularity are useful for assessing how human activities affect these networks. Nestedness often decreases, likely due to habitat loss and a decline in species, affecting overall diversity and interaction abundance, promoting more to generalized networks, while connectance and modularity show variable responses.

Our study demonstrates that the impact of anthropogenic activities on plant-pollinator networks is complex, context-dependent, and varies across different taxa and regions. The findings underscore the importance of considering these factors when designing conservation strategies and policies aimed at mitigating the negative effects of anthropogenic activities on biodiversity within these networks. Further research may be needed to identify additional variables that contribute to the observed heterogeneity and to develop more targeted conservation approaches.

## ACKNOWLEDGEMENTS

We thank Alejandro Restrepo and two anonymous reviewers for their invaluable feedback regarding the manuscript. This work constitutes partial fulfillment of Karla María López-Vázquez's doctorate degree requirements at UAM.

### Funding

This work was supported by the Consejo Nacional de Ciencia y Tecnología, México through a doctoral scholarship to Karla López-Vázquez. The funders had no role in study design, data collection and analysis, decision to publish, or preparation of the manuscript.

### Grant Disclosures

The following grant information was disclosed by the authors:
Consejo Nacional de Ciencia y Tecnología, México.

### Competing Interests

The authors declare that they have no competing interests.

### Author Contributions

- Karla López-Vázquez conceived and designed the experiments, performed the experiments, analyzed the data, prepared figures and/or tables, authored or reviewed drafts of the article, and approved the final draft.
- Carlos Lara conceived and designed the experiments, performed the experiments, analyzed the data, prepared figures and/or tables, authored or reviewed drafts of the article, and approved the final draft.
- Pablo Corcuera conceived and designed the experiments, performed the experiments, analyzed the data, authored or reviewed drafts of the article, and approved the final draft.
- Citlalli Castillo-Guevara conceived and designed the experiments, performed the experiments, analyzed the data, authored or reviewed drafts of the article, and approved the final draft.
- Mariana Cuautle conceived and designed the experiments, performed the experiments, analyzed the data, prepared figures and/or tables, authored or reviewed drafts of the article, and approved the final draft.

## Data Availability

The raw data and the raw data utilized for the meta-analysis are available in the Supplemental Files.

## Supplemental Information

Supplemental information for this article can be found online at http://dx.doi.org/10.7717/peerj.17647#supplemental-information.

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
