# Peer review of "The human touch: a meta-analysis of anthropogenic effects on plant-pollinator interaction networks"

_PeerJ, doi:10.7717/peerj.17647_

## Round 0.1 · original submission · Major Revisions

Dear Dr. Lara,

After this first review round, the reviewers believe your manuscript deserves a second chance to be published. Still, they (and I agree) believe that major improvements are required. As you will see, they have made many suggestions to improve your text. In summary the main issues they raised concern (but are not limited to) the methods (are they reproducible?); more details on the metrics and variables used. Minor improvements throughout the text in all sections. The written English also needs some polishing. New searches for manuscripts using different keywords. A graph to summarize the main results. General description of the pool of manuscripts found during the data gathering. A mixture of processes and patterns in the introduction.

Please consider all suggestions and prepare a rebuttal letter for the next version informing the reviewers those improvements you and your co-authors performed and those that you did not, justifying the reason you did not do the changes. Please do not hesitate to reach me in case you need any assistance or more time to deliver your new manuscript.

Sincerely,
Daniel Silva

Reviewer 1 ·

Basic reporting

In this study, the author made a literature review of anthropogenic effects on pollination networks across the world. Specifically, they searched for papers that reported differences in structural metrics between natural and anthropogenic sites and aimed at identifying which disturbance types led to the strongest metric changes, and the same was made for species abundances. The authors found variation in some network indices across disturbance types, as well as a slight effect of disturbance on pollinator and plant richness. I commend the authors for this contribution to pollination and network ecology, as the effects of anthropogenic disturbance of networks are still poorly understood. While I am mostly positive about the work, some major issues need to be addressed before further consideration.
My biggest concern is the methodology. First, the amount of details given for the literature search is lacking, and in many cases, ambiguous. More care is necessary while making the selection and exclusion criteria explicit for the manuscript to be reproducible. Some of the variables that the authors used (e.g., classification of tropical or temperate environments) have no mention of how they were collected. The climates used, in my opinion, are problems on their own, and I lay out my suggestions in detail below. Pollinator grouping is confusing at best. All in all, more details and precision are necessary throughout.

Second, the description and choice of metrics used need much more work. Authors frequently mistake metrics for network properties, and the metrics themselves have misleading explanations. This needs to be revised and written with more care. In terms of variables included in the study, many factors that would affect the observed outcomes were not considered in the study, such as network size, sampling completeness/effort, whether networks are complete or incomplete, etc. These have been brushed off in the analyses and barely discussed (or not discussed at all). Some properties, for example, are very dependent on network size (connectance, nestedness). Specialization suffers from sampling incompleteness. Modularity is highly dependent on whether more than one functional group (e.g., bees and butterflies) were included in the same network, etc. These variables need to be taken into account, or at the very least acknowledged and discussed. Meanwhile, some analyses are confusing, superfluous, or not well introduced (e.g., the “continuous” variable analysis). Please revise this technical aspect of your manuscript, which would greatly benefit from a rework.
Third, I believe there is a conceptual issue. Although the manuscript tackles networks, the bulk of it (and the most interesting analysis in my opinion, the effects sizes) deals with pollinator and plant communities, with no connection to networks whatsoever. I understand the link between the two, but as far as your analysis goes, half of your paper does not deal with networks. This requires some acknowledgment, and some conceptual adaptation in your introduction, title, and discussion, in my opinion.

Minor and specific issues are listed below:

Introduction

L67-69: I feel this transition towards plant-pollinator interactions (PPI) is a bit abrupt. Within this anthropization context, why are PPI particularly important, and why should they be studied?
L70: Here again, the introduction to networks is quite hasty. PPIs are not just networks; networks are a tool for understanding a species' level of organization about PPI. Give context about what networks are, and how they help answer your research question. I suggest you link it better with L116-120, in which you introduce networks (while giving more context and explanation).
L80: A reference to the literature would be required here.
L89-98: Here you delve into urbanization, but that is just one of the disturbance categories in your work. I suggest that you either give a brief introduction to all of your categories or tone down the focus on urbanization here.
L104-111: Here again with fragmentation. Avoid giving species attention to specific types of disturbances, as it may be read as a bias.
L117: That also requires a reference to the literature.
L119: I agree with you, but you did not specifically measure disturbance intensity, only disturbance categories. Therefore, I would abstain from bringing up this assumption.
L124: I suggest not mentioning the indices’ names (e.g., NODF, Q) here, they are too technical, and this can be done in the introduction.
L130-132: Your manuscript would profit greatly from you laying your hypotheses, as a guide for the reader and also for you. I can tell that you did have them, as you show signs of having initial expectations of the outcome (L480-482), so please state them at the end of the introduction.

Methods

L136: I think stating which authors did the search is unnecessary.
L137: Why was the reason for the search to start in 2010?
L140: I find it very problematic the usage of “such as” here. Are these all the search terms, or just examples? You have to cite precisely all the search terms you used, so I suggest adjusting this sentence. Additionally, why was the reason for restricting the search terms for these specific groups? What about wasps, ants, and other groups? More detail is needed here on the boundaries of the search.
L144: To my understanding, you only selected papers that measured network metrics between a natural site and an anthropized site, is that correct? This is unclear for most of the methods, so I suggest clarifying it here.
L144: Did you include binary, quantitative networks, or both? Very important information that was missing.
L146: To compare pollinator and plant richness, each study must have a comparable sampling effort between the disturbed and undisturbed sites. Was that taken into consideration?
L147: Why three sites? What's the reasoning behind it? Also, at what resolution? Was it three sites per treatment (disturbed and undisturbed) or in total? Many studies work with one site but many small-scale subsites, for example. Needs much more detail.
L150-151: What does this mean? How were disagreements resolved? This sentence is a bit unclear. Also, it is that co-authors were consulted, and the voting part and self-explanatory and thus in my view unnecessary.
L155: I think it’s problematic how the metrics are phrased throughout the methods. There are many other metrics for nested apart from NODF, as well as for modularity and Q and specialization, and H2’. Be careful not to confound a specific concept (e.g., Nestedness) with its metric (e.g., NODF). For example, H2’ measures species interaction overlap between species, and that is a proxy for specialization. Also, please state clearly why you chose specific metrics, and explain them.
L155: Connectance, not connectivity.
L158-160: Consider revising this explanation of nestedness, as it is poorly phrased. Nestedness is a property achieved when the interaction of specialists is a subset of the interactions of generalist species.
L166: As you did for H2’, please state the range of the other metrics.
L166: But why are you exploring this metric? A quick reference (here and for the others) at the end to how and why urbanization might affect it (i.e., your prediction), would be beneficial.
L169-170: Needs a reference for that. Also, not necessarily true, as you may see for networks that are inherently modular and particularly stable. Also, connectance mainly reflects network size, hence why you need to consider sampling effort between works and accounting for network size in your analyses.
L171-173: Exactly, it's mostly a matter of size, less so of other factors. Hence again why size should be accounted for. I truly think this metric is relatively uninformative for you and only confounds the results. Consider removing, or at least clearly stating its limitations and link to community size.
L174-178: Here again, you are mistaking metric for concept. What varies from 0 to 1 is H2’, which is a metric for niche overlap. Also, a high H2’ does not mean a predominance of specialist species, but rather that interactions made by pollinators with plants don’t overlap to a big extent with other pollinators. Hence the network is “specialized”, not the pollinators necessarily.
L180-181: Please revise this affirmation, as it goes the wrong way around. Network specialization is effective specialization, a product of the interactions themselves, and not vice versa. You cannot predict if species are more vulnerable by looking at the result of their interactions, as specialization is a product of what you see and could be driven by factors such as disturbance.
L182: Not one group, but two to several.
L189-192: It’s also a matter of the number of taxa included. Is it a single taxa network (e.g., only bees) or a multi-taxa network? That is quite a determinant of the degree of modularity and should be accounted for in your analyses/review. I suggest stating if it was a criterion or not on your methods to include only single-taxa networks or mixed networks. If you include multi-taxa networks, it should be controlled in your results, just like network size or sampling effort (e.g., via mixed effects models, such as a GLMM).
L201: There is no mention of how these were collected from the papers, i.e., what were the criteria to classify the disturbance in one paper into one category or another. And what about interactions between types of disturbances that can be related (e.g., habitat loss x fragmentation), why were these not considered?
L226: This semi-arid problem brings out a can of worms. First, dry forests are also part of tropical environments, common in South America. Second, it brings the question of what you consider tropical, as this classification was never stated in the methods. Is it anything between the tropical lines, or is it related to climate/biomes? What kind of climates/biomes were found in the studies, and is it worth it to lump them into two “climate” categories (i.e., tropical, temperate)? Over, there is a lot of missing information regarding the environments from which networks were collected, and how this data was assessed. Additionally, I think this division between tropical and temperate should be refined to include the different biome types that you may find (e.g., tropical rainforests, savannas, grasslands, semideciduous forests, etc). One suggestion that could help disentangle this is using climate categories from the Köppen scale, which would be much more informative.
L231-254: Throughout this section, it is not very clear what exactly you are testing. Please specify in detail your dependent and independent variables.
L270: Year of publication and year of sampling are not continuous variables! Rather, they follow a Poisson distribution. Furthermore, there was no previous context to this linear analysis, and it feels very out of place, giving the impression that they were made just because this “continuous” data was sitting there. Assessing, for example, this specific type of disturbance (fragmentation) when your work deals with many more, without giving previous context about it, makes it all very confusing. Also, the year of publication and of sampling (which in my opinion are very redundant) could be used to assess trends in literature, but, as I said, there is no previous mention of these things. I suggest either removing this section entirely or, if you choose to keep it, introducing it better and adapting your objectives and hypotheses.

Results

L284: I suggest mentioning the number of papers per continent.
L293: It’s not necessary to mention again the method that you used in the results.
L296-299: This could be neatly summarized into a boxplot, and would aid in visualizing the extent of the differences.
L312: Just pollinators, or pollinators and plants?
L313-314: This interpretation could be reserved for the discussion.
L323-325: One thing to consider about plants is that they often change significantly once you step into the city, for example, mostly due to ornamentals and invasives. I missed some mention of this in the discussion.
L337-339: Hard to assess the validity of this without a figure of study number per site (which I figure is not many by continent since you only had 16 studies?).
L342-345: This should go to the discussion.
L348: Which former? I think you forgot to write down “temperate” for the 0.77 value.
L351-352: Not necessary to repeat the methods here.
L355-357: See my comment on the methods. Apart from not being continuous, the year of publication/sampling is innocuous for the overarching question of the manuscript as it is now. Same for fragmentation, in the way that your objectives are laid out. In my opinion, other more informative variables other than the year of publication could be included, such as some measure of disturbance intensity, but I understand that this might not be possible due to differing methodologies in the papers.
Table 2: Pollinator grouping is confusing. This needs to be addressed here and in the methods (i.e., how the classification and division were made). For instance: “insects” and “bees and others” both have bees in it. “General” refers to insects even though there is an “insects” category”! Please correct these classifications and be more transparent about it.
Figure 3:
1- This figure is nice but can be greatly improved. For instance, dividing plants and pollinators by color blocks, or adding disturbance type to the figure somehow (by colors, or some way else). It seems a bit crude as it is.
2 - None of the references are in the cited literature list.
3 – Why is Weiner 2011 divided into 5 networks? There is no reference to divisions like this in the methods.

Discussion

General issue: Although part of your analysis was comparing pollinator groups and different continents, there is mention of it in the discussion.
L364: “the” plant-pollinator network? Or networks in general?
L365: On “biodiversity” sounds a bit far-fetched. At best it helps understand the dynamics of mutualistic interactions and the vulnerability of floral visitors.
L377-379: This digresses and is not necessarily important for your discussion
L381: only on interaction networks? Half of your analysis deals not with networks, but with plant and pollinator communities and abundances only. See the major issue regarding this conceptual issue throughout the paper.
L391-393: This sentence brings two different and non-connected ideas. Please rewrite it accordingly.
L399: No need to repeat the metrics here.
L402: Nestedness. Also, as stated at the beginning of the review, other factors could be affecting nestedness but were not taken into account for the analysis. Please include them and/or acknowledge them in your discussion.
L406: “limitations” is not the best term here. It implies a negative judgment of the behavioral and morphological traits of some species. Try specialization instead, or something in this direction.
L411-415: So, in a nutshell, network size is a highly relevant factor. This metric should be included in the analysis.
L421-423: requires a reference from the literature.
L461-464: You cannot state this when your analysis says otherwise. Also, there was no buildup to this in your introduction, as I mentioned before, and the analysis requires corrections. Please adapt accordingly.
L467-474: Due to the problems related to the climate grouping, this will have to be adapted accordingly.
L469-471: I highly doubt this is the cause for this bias. Although most of the biodiversity is indeed found in the tropics, most networks so far have been sampled in the temperate zones, so that is not a good argument.
L486: This is a dangerous and erroneous interpretation. So we should not protect not well-connected communities? These are likely the most disturbed and therefore exactly the ones that would require attention, no?

Experimental design

no comment

Validity of the findings

no comment

·

Basic reporting

I believe that the background is clear and explains the role of this work in the field of anthropization effects on pollination interactions. However, it is necessary to complement the introduction with more references, possibly including more recent works. Additionally, reorganizing the citations in some paragraphs would make it easier to consult the papers of interest. For more details, I added some comments in the introduction in the pdf.

Experimental design

I believe that the authors nedd to review the lines 150-152 to avoid a bias in their method, I added a comment in the pdf. References are needed for the network metrics described in the methodology, especially when explaining their significance and interpretation.

The statistical methods appear appropriate and were well-explained to address the research question.

In Figure 1, you mentioned articles obtained from other sources, but this is not clear in the methodology.

Validity of the findings

I believe the first phrase in the discussion needs to be more specific and less general. It is not clear in the work how identifying changes in the structure of pollination networks can help anticipate changes in biodiversity.

The authors should consider including citations in the discussion, especially when referencing results from the analyzed articles. Additionally, certain conclusions or explanations need to be supported by references.

Although the discussion is well-constructed, I believe that the authors need to consider adding more citations in some parts. Additionally, some statements may require caution to avoid misinterpretation, especially considering the limitations of the study. I have made some comments in the pdf.

Reviewer 3 ·

Basic reporting

Major comments:

In this paper the authors carried out a systematic literature review to understand the effect of different types of anthropogenic land use on plant-pollinator interaction network metrics (specialization, connectance, nestedness and modularity). Furthermore, they carried out a meta-analysis and assessed whether the size of the effect on pollinator and plant richness in disturbed and undisturbed areas. They also verified whether this effectvaried depending on the study interaction group (plants or pollinators), climate where it was carried out, anthropogenic activity and continent.

The topic of the study is very relevant, however the text presents several flaws and some inconsistencies that are specified in detail below in “Specific/minor comments”. The writing lacks clarity, with paragraphs and sentences in the introduction and discussion that are not logically connected, hindering the flow of reading. Additionally, there are some theoretical inconsistencies. Therefore, for this study to be accepted, a thorough review of the writing and structuring of the text with a more logical sequence in the introduction and discussion is necessary.

Regarding the methodology, I also think it is important to consider some points, such as carrying out a new search using all types of land use evaluated in the study and not just “fragmentation” and “land use change”. I also think it is important to consider the question I raised about the category “land use change” since all others assessed are a type of land use change. I also think it is important to improve the graphical part of this study. My suggestion is that you make a graphic summary with illustrations about the main results found. However, what concerns me most are some inconsistencies in the results that I described in more detail in the specific comments that need to be checked. I hope to have contributed to improving this study and I believe that after all these suggested adjustments, this research could be considered for publication.

Experimental design

Nothing to add. It has already been described in the "basic reports" section

Validity of the findings

Nothing to add. It has already been described in the "basic reports" section

Additional comments

Specific/minor comments:
L64-67: In this part you are mixing the consequences of anthropization (e.g. changes in land use, urbanization, infrastructure development) with the environmental impacts caused causados (e.g. deforestation, agriculture, mining, and pollution) setting everything as an example of economic and political challenges. It's confusing and needs to be improved. My suggestion is that you first write a little about the main types of land use conversion caused by the anthropization process and then describe the environmental consequences. Then you could finish the sentence by describing that all these factors lead to a series of economic, political, and mainly environmental challenges (which is the focus of your study and should be mentioned).

L68-L69: You really need to structure this entire paragraph. Here in these lines you have placed the main theme of your study without any contextualization. My suggestion is that after you list the impacts linked to anthropization (as mentioned above) you describe the general impacts caused to biodiversity (example, biotic homogenization, decrease in species diversity, loss of specialist species, decrease in functional diversity, among others). After this contextualization, you introduce your main theme by saying that these changes in land use can also alter interactions between species, for example, between pollinators and plants. Remember that a good introduction makes all the difference in the study, the reader needs to be curious when reading your introduction wanting to know more about your research, and this first paragraph does not convey that idea.

L74: The terms “disturbance gradients” and “land use modification” are not correct in this context because the disturbance gradient can be applied in any of the examples you describe such as, urbanization, habitat loss, and fragmentation. Furthermore, they are all a type of land use modification. Do you understand the difference?

L79: Replace“striking” for “is reported in many studies” Pay attention to more scientific language.

L-82-84: Describe in more detail about why these ecosystems affect species differently, focusing on the landscape variation of these modified environments, for example, how different an urban environment is from a rural environment, and describing more about the biological characteristics of the species that are also responsible for these different responses.

L85-88: I don't understand why you started writing about cultivated land here if at the beginning of the paragraph you are describing pollinators' responses to habitat changes in general. Improve the context so as not to leave the reader lost!

L89-98: It was strange to write in the introduction only one of the changes in land use, in this case urbanization, considering that in your study you also evaluated agriculture, fragmentation, land use changes and fires. It would be more recommended that you write a little about the impact of each of them to justify in your objectives because you defined them, or it is better if you write about the impact of land use changes in general. Furthermore, in this paragraph you describe the impact of applying pesticides and herbicides in urban areas. This impact is much more related to agricultural areas that intensively use more pesticides. Therefore, I do not recommend that you associate this discussion with urban areas, which have other factors that can have a much greater impact on biodiversity.

L99-104: The introduction is completely unsequenced, this part should be at the beginning of your introduction after saying that anthropogenic changes can lead to changes in interactions, and then you could describe that this could result in the loss of ecosystem services due to changes in the process reproductive relationship between pollinators and plants.

L104-111: Same L89-98.

L136: This information about who conducted the literature search should not be there. Normally this information is in the section“author statement”.

L137: Why did you limit searches after 2010? You need to justify this here. Furthermore, in meta-analyses it is important to report the last date of the search.

L139: You should have included the other types of land use evaluated in your study in the search words, but you only used “fragmentation” and “land use change” this could contribute to bias in your study. You should do a new search including “agriculture” “urbanization” and “fires”. Furthermore, land use change would not be a category because all other categories are a type of land use change.

L150: You must specify all exclusion criteria when doing a systematic review or meta-analysis. This should be clear to anyone reading your research.

L152: This decision is arbitrary that's why is important to establish clear exclusion criteria.

L226: Few how many? Need to report.

L287-L293: This part of the results is not very informative, it is not linked to your main objectives and because you mention the effect of the metrics it is confusing with the part you did in the meta-analysis. Furthermore, this information is already described in table 1, there is no need to repeat it here.

L284-L285: Why was livestock farming one of the most prominently if it was one of the least evaluated categories as shown in table 1?

319-348: It would be interesting to make a summary figure to improve the visualization of the relationships found here.

L363: It is recommended to summarize the most relevant results found in your study in the first paragraph of the discussion. I think this will help you discuss more clearly. In general, I also found the discussion confusing and I think that, like the introduction, it needs to be better structured.

L469-474: I think this information needs to be reviewed, normally this pattern is the opposite, most of these studies are carried out in temperate regions, with the tropical region being least studied despite its greater diversity. Below I have listed some of the works that talk about this:

Plant-pollinator networks in the tropics: a review. Ecological networks in the
tropics, Springer, 73-91.

Diverse urban pollinators and where to find them. BIOLOGICAL CONSERVATION, v. 281, p. 110036, 2023

---

## Round 0.2 · Minor Revisions

Dear Dr. Lara,

The reviewers believe there were significant improvements concerning the previous version of your manuscript. I believe that after a small change suggested by one of the reviewers, the manuscript will be accepted for publication.

Sincerely,
Daniel Silva

Reviewer 1 ·

Basic reporting

I think the authors did a very good job of incorporating the suggestions and corrections in this new version. I don't have any major comments left, except regarding the Köppen scale issue:

-The authors mention improving the classifications of the climates found using the scale, but I don't see its effects in the results. In Figure 5, for example, I still see "Dry", "Tropical" and "mid-latitude Climates". This to me is still entirely unclear (what is "tropical"? There are several tropical regions that are dry) and does not correspond to the Köppen scale in any sense. Please adjust the results accordingly.

Experimental design

no comment

Validity of the findings

no comment

Additional comments

no comment

·

Basic reporting

I believe the corrections made to the manuscript were well-executed, with the authors incorporating all of the reviewers' suggestions to enhance the quality of the manuscript. Both the introduction and discussion underwent substantial changes to accommodate the suggestions, and these changes were well-implemented. The methods section was augmented to improve clarity and avoid interpretation issues. Additionally, the figures were revised and now offer better clarity. All of the changes made to the manuscript contribute to its overall high quality.

Experimental design

All changes in the methods section were well executed and contribute to better alignment with the manuscript's objectives.

Validity of the findings

All the changes made in the discussion section provided a clear context for interpreting the results. The inclusion of new references further validated the findings.

Reviewer 3 ·

Basic reporting

The authors have addressed all modifications proposed during my review in favor of improving the study, thus I am in agreement with the publication of this article.

Experimental design

The authors have addressed all modifications proposed during my review in favor of improving the study, thus I am in agreement with the publication of this article.

Validity of the findings

The authors have addressed all modifications proposed during my review in favor of improving the study, thus I am in agreement with the publication of this article.

Additional comments

The authors have addressed all modifications proposed during my review in favor of improving the study, thus I am in agreement with the publication of this article.

---

## Round 0.3 · accepted · Accept

Dear Dr. Lara,

I am pleased to accept your manuscript for publication in PeerJ.

Sincerely,
Daniel Silva